# The Effect of Capping Agents on Gold Nanostar Stability, Functionalization, and Colorimetric Biosensing Capability

**DOI:** 10.3390/nano12142470

**Published:** 2022-07-19

**Authors:** Tozivepi Aaron Munyayi, Barend Christiaan Vorster, Danielle Wingrove Mulder

**Affiliations:** Human Metabolomics, North-West University, Potchefstroom Campus, Potchefstroom 2531, South Africa; 26718944@g.nwu.ac.za

**Keywords:** capping agents, biosensor, colorimetric, gold nanostar, physico-chemical

## Abstract

Capping agents (organic ligands, polymers, and surfactants) are pivotal for stabilizing nanoparticles; however, they may influence the surface chemistry, as well as the physico-chemical and biological characteristics, of gold nanostar (AuNS)-based biosensors. In this study, we proved that various capping agents affected capped and bioconjugated AuNS stability, functionality, biocatalysis, and colorimetric readouts. Capped and bioconjugated AuNSs were applied as localized surface plasmon resonance (LSPR)-based H_2_O_2_ sensors using glucose oxidase (GOx) as a model enzyme. Furthermore, our analyses revealed that the choice of capping agent influenced the properties of the AuNSs, their stability, and their downstream applications. Our analyses provide new insights into factors governing the choice of capping agents for gold nanostars and their influences on downstream applications with conjugated enzymes in confined environments.

## 1. Introduction

Gold nanoparticles (AuNPs) are attractive for use in nanomedicine due to their facile synthetic methods, sensitivity, specificity, possibilities for tailoring their charge, hydrophilicity, and functionality through surface chemistries [1,2,3]. AuNPs come in a variety of morphologies that are tuneable for biomedical purposes. A single nanomorphology may be used in therapy, diagnostics, or even theranostics [4]. An example of this includes the use of nanorods as drug delivery carriers and as cell-imaging and photothermal therapies for cancer [5]. Due to their inert nature and biocompatibility, gold nanoshells are used mainly in drug delivery, gene delivery, photothermal therapy, cancer cell targeting and treatment, and medical biosensing [6]. Gold nanostars are best-suited for biosensing applications due to their strong localized surface plasmon resonance (LSPR), which maximizes near the sharp tips of the particles upon excitation from a selected electromagnetic frequency [7].

Gold nanostars (AuNSs) are an auspicious morphology with arms protruding from a more or less spherical core. These arms enhance plasmonic reactions based on the lightning rod effect, which has the highest enhancement factors in localized surface plasmon resonance (LSPR) biosensing [8,9]. Surface plasmon resonance can be tailored by controlling a nanostar’s arm density, length, and physicochemical surface structure without altering the overall dimensions [7,10]. This ultimately redefines the limits of AuNS detection by presenting a signal-generating mechanism that induces a more significant signal when the analyte is less concentrated [10,11]. In biological applications, aqueous synthesis is commonly used, although in some instances, AuNSs may be synthesized in organic media [12,13]. This helps to achieve better control over the physical and chemical properties and homogeneity of the colloidal solution [13]. Changes in essential parameters, such as the synthesis method, capping agents, pH, temperature, concentration, and environment, greatly influence the quality and quantity of the synthesized AuNSs, as well as their characterization and applications [14].

Capping agents are usually utilized as stabilizers when creating AuNSs, with these agents being involved in modulating surface chemistry, morphology, and size distribution, as well as preventing coalescence and aiding in the formation of complex structures with metallic ions in precursor salts [15,16,17]. Capping agents influence nanoparticle shapes, while the capping agent length, size, ω-functionalities, and chemical nature influence the shelf-life and colloidal stability of AuNSs in dispersions [18]. Capping agents play a key role in altering the surface charge, furthermore acting as bioconjugation scaffolds [19,20,21]. Capping agents—enzyme conjugates—are essential considerations in biosensor preparations. In many cases, biocompatible capping agents are commonly used [1,2,22,23,24]. Capping agents may improve enzyme stability, shield enzymes from undesired interactions, and prolong shelf-life [23,25,26]. Such shielding effects rely on the physicochemical properties of the capping agents, such as the hydrophilicity, chain length, and architecture [25,26,27,28,29]. Capping agents can be categorized broadly as natural or synthetic surfactants, small ligands, polymers, dendrimers, cyclodextrins, or polysaccharides [3,27,30]. Polymeric capping agents are usually very effective due to the entropic effect associated with multiple bindings and the steric stabilization provided by the macromolecule [27,30,31]. Commonly used polymers are polyvinylpyrrolidone, polyvinyl acetate, polyethylene glycol (either modified or derivatives), poly(acrylic acid) (either modified or not with additional binding agents such as amines), copolymers based on poly(maleic acid), and polyethyleneimine [3,27,31,32].

In principle, a strongly adsorbing capping agent provides better colloidal stabilization to AuNSs, but in some instances, it is necessary to use a weaker capping agent in the synthesis depending on the downstream applications [3,33,34]. The adsorption of capping agents on AuNS surfaces can be through chemisorption, electrostatic interaction, or hydrophobic interaction [30]. However, there may be a pressing need for using eco-friendly (green) capping agents to secure the biological system and the environment, as some are nonbiodegradable and toxic [31,32].

The following study is conducted based on the ambiguity of why researchers choose specific capping agents over others. A one-pot, silver-assisted nanostar synthesis method using 4-(2-Hydroxyethyl)-1-piperazineethanesulfonic acid buffer (HEPES) is the synthesis method of choice. Silver nitrate is used as a shape-directing agent to assist the HEPES in obtaining monodispersed, multibranched gold nanostars (AuNSs). The effects of various capping agents on AuNS stability are investigated in environmental conditions used in biosensing work. Additionally, the capping agents’ influences on the bioconjugate and enzyme activity using GOx as a model enzyme are evaluated. Furthermore, how the capping agents influence the colorimetric potential of the AuNS bioconjugates is assessed based on hydrogen peroxide detection. The biocatalytic growth of AuNS is the mechanism of detection through surface-coating with silver (Ag) after its reduction with hydrogen peroxide, as proposed by the following equations [9,33]:(1)$$\beta -d-\mathrm{Glucose}+O_{2}+H_{2}O\;\overset{\mathrm{GOx}}{\to }\;d-\mathrm{Gluconic}\;\mathrm{acid}+H_{2}O_{2}$$
(2)$$3H_{2}O_{2}+{\mathrm{Ag}}^{+}\overset{\mathrm{AuNS}}{\to }3H_{2}O+\mathrm{Ag}\;+\frac{3}{2}{\;O}_{2}$$

## 2. Materials and Methods

### 2.1. Materials

4-(2-Hydroxyethyl)-1-piperazineethanesulfonic acid (HEPES), gold (III) chloride hydrate (HAuCl_4_·4H_2_O), glucose oxidase (GOx), hydrogen peroxide (H_2_O_2_), silver nitrate (AgNO_3_), polyethylene glycol (PEG 8000), polyethylene oxide (PEO 100,000), sodium chloride (NaCl), polyvinylpyrrolidone (PVP 10,000), cetyltrimethylammonium bromide (CTAB), 11-mercaptoundecanoic acid (MUA), polyethylenimine (PEI), trisodium citrate, gelatin chitosan, glucose, 3′30–dithiobis sulfosuccinimidyl propionate (DTSSP), and sodium hydroxide (NaOH) were purchased from Sigma-Aldrich (Johannesburg, South Africa). Ham’s F-12K (Kaighn’s) cell culture medium supplemented with 10% fetal bovine serum was purchased from Thermo Fisher Scientific (Johannesburg, South Africa). Blank urine was purchased from Industrial Analytical (Pty) Ltd. Lastly, ultrapure water (ddH_2_O) was pre-prepared with a Milli-Q ultrapure system (18.2 MΩ cm^−1^).

### 2.2. Nanostar Synthesis

The nanostars were synthesized according to Xie et al. and Mulder et al. [10,34]. Briefly, 3 mL H_2_O was added to 2 mL 100 mM HEPES buffer (pH 7.4), followed by the addition of 20 µL 50 mM HAuCl_4·_H_2_O and, lastly, 4 µL 1 mM AgNO_3_. The solution turned blue after gentle end-to-end inversion mixing and incubation at room temperature for 30 min.

### 2.3. Nanostar Capping Agents (AuNS–Capping Agent)

Capping of the nanostars was achieved by adding ranges of PEO, PVP, PEG, CTAB, MUA, PEI, trisodium citrate, gelatin, and chitosan concentrations, followed by salt stress testing as outlined below. Optimal concentrations were selected based on lack, or limited degradation, after visual and spectrophotometric assessment. The nanostars were incubated for one hour for each of the selected concentrations, after which they were centrifuged at 4180× *g* for 35 min. Following this, the soft pellets were washed and resuspended in 500 µL ddH_2_O. The clean-up process was repeated twice.

### 2.4. Enzyme Bioconjugation (AuNS–Capping Agent–GOx)

The PVP-, PEO-, and PEG-capped AuNSs were bioconjugated with glucose oxidase (GOx) for further investigation. This was to determine if the capping agent affected the enzyme bioconjugation process and biosensor colorimetric reactions. A method suggested by Filbrun et al. with minor modifications was followed to attach GOx to the AuNSs using DTSSP [35]. Briefly, capped AuNSs were synthesized as described above but were resuspended in 500 µL 100 mM HEPES (pH 6.9) as opposed to ddH_2_O in the final step. Four batches of 500 µL each were then pooled, making a total volume of 2 mL. An amount of 100 µL 5 mM DTSSP was then added to the solution, followed by 150 µL 2 mg/mL GOx. The solution was then left in the fridge for 2 h, allowing for AuNS–GOx attachment. Excess DTSSP and GOx were removed through 2 cycles of centrifugation at 4180× *g* for 35 min and resuspension in 500 µL ddH_2_O.

The stability of the GOx-bioconjugated AuNSs (AuNS-GOx, AuNS-PVP-GOx, AuNS-PEO-GOx, and AuNS-PEG-GOx) was then confirmed using a stress testing protocol, as described below.

### 2.5. Characterization

Characterization of the nanostars was performed with spectroscopy and high-resolution transmission electron microscopy (HR-TEM) analyses (JEOL, Freising, Germany). Spectral scanning (400–900 nm) was conducted with an HT Synergy (BioTEK) microplate reader (Agilent Technologies, Santa Clara, CA, USA). High-resolution transmission electron microscopy (HR-TEM) analysis was performed with a Tecnai F20 transmission electron microscope at an acceleration voltage of 200 kV (JEOL, Freising, Germany). The capped AuNS samples were spotted onto copper grids (Agar Scientific), followed by air-drying before imaging, whereas the AuNS bioconjugates needed staining for enzyme visualization. This was performed by spotting the samples onto copper grids, staining them with 1% silver nitrate solution, and allowing them to air-dry before imaging [9]. The particle core diameters and arm counts were estimated by averaging a total count of one hundred AuNSs in different grid regions using ImageJ software (ImageJ bundled with 64-bit Java 1.8.0_172, University of Wisconsin at Madison, Madison, WI, USA).

In addition, successful PVP-, PEO-, and PEG-capping and enzyme bioconjugation of the nanostars were demonstrated with nuclear magnetic resonance (NMR) and agarose gel electrophoresis. NMR was performed at 500 MHz with a Bruker Avance III HD NMR spectrometer equipped with a triple resonance inverse (TXI) 1H (15N, 13C) probe head (Bruker, Billerica, MA, USA). Agarose gel electrophoresis of capped and AuNS bioconjugates was carried out using a Baygene BG-power Vacutec electrophoresis gel apparatus (Vacutec, Johannesburg, South Africa). The electrophoresis was conducted using 0.5% agarose and 0.5 × Tris borate EDTA buffer (TBE buffer) at pH 8. The samples were prepared using 32 µL samples of nanostars mixed with 4 µL 80% glycerol and were run at 40 V for 45 min. Gels of AuNS bioconjugates were stained with 25 mL 0.25% (*w*/*v*) Coomassie blue for 4 h, followed by rinsing with a destaining solution (90% *w*/*v* isopropanol and 10% *w*/*v* glacial acetic acid), after which the protein bands were visible. Gels were kept in ddH_2_O until the gel images were captured and transferred to a computer.

### 2.6. Stability Assessment for Capped and AuNS Bioconjugates

The stability during storage and various possible analytical assay matrices of capped and AuNS bioconjugates were assessed. The storage conditions were at both room temperature and 4 °C over 96 h. The analytical assay conditions assessed included various environments (salt, urine, serum, and supplemented cell culture medium) at room temperature for 3 h. A change in the LSPR band OD_max_ of greater than 30% was considered unstable using UV-vis (BioTEK, Agilent Technologies, Santa Clara, CA, USA) [36].

### 2.7. Effect of the Capping Agent on the Plasmonic Properties of AuNSs in Bioassays

The effect of the capping agents on the plasmonic properties of AuNSs when used as chemical sensors in bioassays was assessed first by the incubation of capped AuNSs with hydrogen peroxide, followed by unconjugated and conjugated GOx-generated hydrogen peroxide assays.

For the hydrogen peroxide assay, reagents were added and pipette-mixed in the following order: 20 µL 10 mM Tris buffer (pH = 8), 15 µL capped AuNSs, 0–20 µL 50 mM hydrogen peroxide in increments of 2 µL, 2 µL 10 mM silver nitrate, and 20 µL 500 mM NaOH. As an initial step, a volume of deionized water was added that ensured a final volume of 200 µL. The reagents were incubated for 5 min at room temperature before spectrophotometric measurement [37].

The unconjugated GOx assay was performed as above, but the hydrogen peroxide was replaced with 5 µL 2 mg/mL GOx and 0–20 µL 1.65 mM glucose in increments of 5 µL, followed by 15 min of incubation at 37 °C. Then, a detection solution consisting of 2 µL 10 mM silver nitrate and 20 µL 500 mM NaOH was added and incubated for another 1 min, after which spectral readings were obtained [37]. The conjugated GOx assay was performed as above, with the exception that the GOx addition was omitted, and 0–20 µL 2 mM glucose was added in increments of 5 µL.

## 3. Results

Even though spherical nanoparticles have been capped with CTAB, MUA, PEI, trisodium citrate, gelatin, and chitosan, attempts to reproduce this led to degradation of the AuNSs [38,39]. These agents were subsequently excluded from further investigation. The optimal concentrations of PVP, PEO, and PEG solutions for capping 5 mL AuNS suspensions were 600 µL 25 mM PVP, 250 µL 0.25 mM PEO, and 500 µL 35 mM PEG, respectively.

The UV-vis spectral scans, NMR spectra, and electrophoretic migration of capped (left column) and bioconjugated (right column) AuNSs are presented in Figure 1. The absorbance spectra (A and D) showed a small shoulder peak at around 545 nm and a more intense, red-shifted, prominent peak at approximately 635 nm. AuNS and AuNS-PEO achieved the best spectral resolutions of the two LSPRs, followed by AuNS-PVP and AuNS-PEG. A mild loss in spectral resolution was observed for all the GOx-bioconjugated nanostars except uncapped AuNS-GOx, which was almost identical to AuNS. The change in the ^1^H-NMR spectra when compared to the AuNS spectrum hinted toward the presence of the capping agents (B) and GOx (E) in the samples and agrees with previously reported results [10,37]. The change in the electrophoretic pattern of the capped AuNSs compared to AuNS (C), as well as between the capped AuNSs and GOx-bioconjugated AuNS (F), indicated a change in the charge or size of the nanostars after capping and bioconjugation. The absence of the GOx band (F) in the bioconjugated AuNS lanes suggested that GOx was successfully conjugated to nanostars. All the nanostars migrated toward the anode, with migration distance being indirectly proportional to capping agent polydispersity. The presence of a five-membered ring moiety and a long-chain structure in PVP probably limited its migration in comparison to the linear chemical structures of PEO and PEG. As expected, the attachment of GOx to nanostars decreased migration when compared to unattached GOx.

TEM images of capped (top row) and bioconjugated (bottom row) AuNSs are presented in Figure 2. AuNS (Figure 2A), AuNS-PVP (Figure 2B), and AuNS-PEG (Figure 2D) particles were moderately monodispersed, with an estimated average size ranging from 40 to 42 nm. Most AuNS-PEO particles (Figure 2C) were cleaved during the capping process, with resultant tripod or tetrapod appearances and an average diameter of 25 nm. AuNS-PVP had a lower tendency toward aggregation, and most nanostars had a spiked morphology with 12 arms, of which approximately nine protruded around the nanostar and three protruded outwards. When comparing the AuNS bioconjugates, AuNS-PVP-GOx (Figure 2E) exhibited better monodispersity, with a larger percentage of nanostars having a spikey morphology relative to AuNS-PEO-GOx (Figure 2F) and AuNS-PEG-GOx (Figure 2G). Unexpectedly, a protein layer could not be visualized after 1% silver nitrate staining of AuNS-PVP-Gox despite ample evidence of successful functionalization, as presented in Figure 1. The layer was, however, visible when AuNS-PEO-GOx and AuNS-PEG-GOx were stained, and it was visualized as small dark spots of approximately 3.0 nm around the nanostar bioconjugates, which is in line with previously reported observations [40,41]. The size distribution plots for the AuNS-PVP-Gox, AuNS-PEO-GOx and AuNS-PEG-GOx are in contained in the Appendix A. The AuNS-GOx sample contained mostly aggregated spheres. These nanoparticles were not stable, as was evident from changes in the absorbance spectrum and color of the solution within 30 min after conjugation (Appendix A. 

The stability of the nanostars in various matrices (left column) and the short-term stability at both room temperature and 4 °C (second and third column) are presented in Figure 3. AuNS-PEO and AuNS-PVP maintained stability in all the tested matrices (Figure 3A–D), with a slight blue-shift in the spectrum relative to controls in some instances. AuNS-PEG was stable in the salt only, while AuNS was unstable in all the matrices, as shown by a relative change in the longitudinal resonance OD_max_ of greater than 30% after 3 h of incubation. AuNS-PVP and, to a mildly lesser extent, AuNS-PEO were stable for 96 h at both room temperature and 4 °C (Figure 3F–K). Uncapped AuNSs exhibited stability loss at room temperature, as seen by the blue-shift in the longitudinal LSPR peak, as well as in the visible color change within 48 h of synthesis (Figure 3E). It did maintain stability at 4 °C over the 96 h period (Figure 3I), which was in contrast to AuNS-PEG, which was only stable up to 72 h at both room temperature and 4 °C (Figure 3H,L).

In comparison, the AuNS bioconjugates had minimal loss of stability, demonstrating that enzyme conjugation could induce extra stability in the AuNSs (Figure 4A–D). Contrarily, conjugation had less of an effect on short-term stability, with AuNS-PVP-GOx being the most stable, followed by AuNS-PEO-GOx and AuNS-PEG-GOx. Therefore, the results suggested that the choice of capping agents affects the shelf-life, stability, and structural integrity of nanoparticles in suspension. Overall, 4 °C was the optimal storage condition for the AuNS–capping agent and AuNS–capping agent–GOx combinations.

A conjugated GOx assay was used to assess the efficacy of the various bioconjugates to act as chemical sensors of biocatalytically produced hydrogen peroxide. The results of the assay are presented in Figure 5. The results of the hydrogen peroxide and unconjugated GOx assays are presented in the Appendix A. As expected from the stability studies, AuNS-GOx was an inefficient sensor of hydrogen peroxide, as evident from the lack of correlation between the glucose concentration and the spectral shift (Figure 5B). The longitudinal LSPRs of AuNS-PVP-GOx (Figure 5C) and AuNS-PEG-GOx (Figure 5G) blue-shifted, with the degree of the shift correlating to the added glucose concentration. Only one peak was observed in the AuNS-PEO-GOx assay that was also blue-shifted with increasing glucose concentrations (Figure 5E). AuNS-PEO-GOx and AuNS-PEG-GOx had an immediate color change when the detection solution was added, including a reaction at 0 mM glucose. AuNS-PVP-GOx maintained its initial color at 0 mM glucose and needed 5 min of incubation for visual and spectrophotometric changes to develop at increasing glucose concentrations. AuNS-PVP-GOx outperformed both AuNS-PEO-GOx and AuNS-PEG-GOx as a chemical sensor of biocatalytically produced hydrogen peroxide, as evident from its R^2^, which approached one. The color of AuNS-PVP-GOx was also the most vivid and changed from blue to purple, whereas the other two capping agents produced mostly varying shades of purple.

Table 1 is a summary of the results found in this study and possible downstream applications of the bioconjugates.

## 4. Discussion

In summary, a plethora of the literature has focused on the synthesis of capped AuNSs. However, limited efforts have been devoted to conceptualizing the role played by capping agents in controlling the interface of AuNSs with their environment, and a clear structure–function relationship remains elusive. In this study, AuNSs synthesized through the chemical reduction of gold salt by HEPES showed traces of HEPES (a zwitterion) on the ^1^H-NMR spectra (Figure 1B,E). The AuNSs had two localized surface plasmon resonance (LSPR) modes with peaks at approximately 545 nm and 635 nm, corresponding to the plasmonic resonance of the core (transverse resonance) and branches (longitudinal resonance), respectively, as has been previously reviewed in the literature [42,43,44]. HEPES, which was present in all the ^1^H-NMR spectra, has previously been reported as both a shape-directing and a strong capping agent in nanostar synthesis and is likely incorporated within the nanostar lattice or bound to the surface [34,45]. The interactions between HEPES functional groups (sulfonate, amine, and hydroxyl) and AuNSs can be ion- or dipole-induced dipoles. These molecular interactions are heavily influenced by pH fluctuations, electron density, and capping agent binding chemistries, as revealed by Xi et al. [46]. Quintanilla et al. suggested that the larger the capping agent’s molecular weight, the more catalytically active and stable the polycrystalline nanoparticles, which is in line with our findings [47]. In addition to this, our results indicated that molecular weight and chemical moieties influenced nanoparticle stability and homogeneity. PVP, PEO, and PEG are vinyl polymers with different side groups. The larger side groups on PVP could reduce segmental mobility and hydrogen bond density near the interface and lead to lower adhesion energy, less fragility, and degradation, hence increasing biosensor stability, a phenomenon previously reported by Lucius et al. [48]. Strong hydrogen bonds at the interface influenced major chemical changes, such as chain oxidation and chain scission, leading to a reduction in the molecular weight and degree of polymerization of a polymer, as in the case of PEO- and PEG-capped AuNS.

The use of PVP as a capping agent in the GOx bioassay resulted in vivid color changes in response to varied hydrogen peroxide concentrations for both bioconjugated and capped particles despite its slower response time. The PEO and PEG bioconjugates produced fast results, regardless of their instability relative to the PVP bioconjugate, and their colors resulted in a change in color intensity as opposed to a color range. Our results demonstrated that tailoring (capping) polymer functionality could improve bioconjugate activity and minimize biosensor instability. These observations suggested that the choice of capping agent has major implications in laboratory-based and point-of-care testing diagnostic applications.

Capping agents introduce a protective layer on the AuNS surface, hence introducing two major interfaces, namely the AuNS–capping agent interface and the AuNS–capping agent–protein interface. There is no clear demarcation of these interfaces, and they seem intermixed in a transition zone, the AuNS–ligand interface. Similar surface chemistry analyses have been revealed by Grubbs and Campisi [17,49]. Electronic changes at the AuNS–ligand interphase unavoidably affect the catalytic behavior of capped nanoparticles and bioconjugates, introducing either a poisoning or promoting effect. A similar effect was revealed by Tsunoyama et al. for PVP-capped Au nanoparticles [50]. The inherent complexity of AuNS–capping agent–protein interactions also contributes to the challenge of deducing general conclusions on the best choice of a capping agent. Overall, the choices of a synthesis method and a suitable capping agent should be key considerations in bioassay design, seeing that they affect the stability, surface chemistry, catalytic response, and selectivity of AuNS sensors.

## 5. Conclusions

The AuNSs resulting from this facile and repeatable HEPES-mediated, one-pot synthesis protocol were approximately 38 ± 4 nm in diameter, multibranched, monodispersed, and homogeneous to a great extent. The contribution of our work is two-fold. Firstly, we demonstrated that capping agents affected AuNS stability, functionalization, and colorimetric biosensing capabilities. For polymeric capping agents, we revealed that fewer hydrogen-bond-forming groups in a polymer side group could reduce the hydrogen bond density near the interface, hence increasing biosensor stability. Secondly, we showed that the confinement effects induced by free surfaces and substrates in supported polymer-capped and conjugated biosensors could improve the bioconjugate activity and biosensor stability. Conclusively, PVP capping proved to be optimal for HEPES-mediated AuNS synthesis, and the fabricated glucose biosensor proved to be a rapid colorimetric assay that may be a great candidate for potential clinical diagnosis, research, and development applications useful in resource-constrained regions.

## Figures and Tables

**Figure 1 nanomaterials-12-02470-f001:**
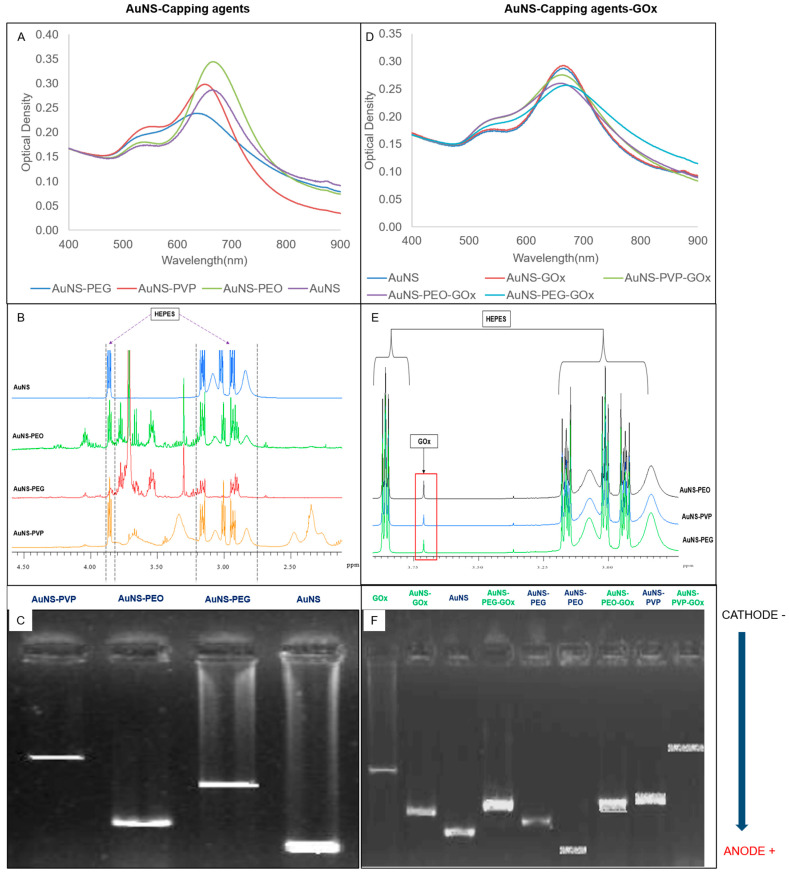
(**A**–**C**) Effects of capping agents on AuNSs using (**A**) UV-vis spectrometry, (**B**) nuclear magnetic resonance (NMR), and (**C**) gel electrophoresis. (**D**–**F**) AuNS capping agents functionalized with glucose oxidase using (**D**) UV-vis spectrometry, (**E**) NMR, and (**F**) gel electrophoresis.

**Figure 2 nanomaterials-12-02470-f002:**
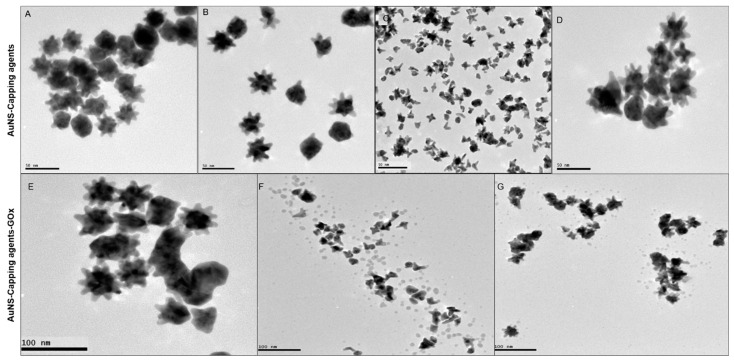
Representative HR-TEM images of the gold nanostar morphologies: (**A**) AuNS; (**B**) AuNS-PVP; (**C**) AuNS-PEO; (**D**) AuNS-PEG; (**E**) AuNS-PVP-GOx; (**F**) AuNS-PEO-GOx; and (**G**) AuNS-PEG-GOx.

**Figure 3 nanomaterials-12-02470-f003:**
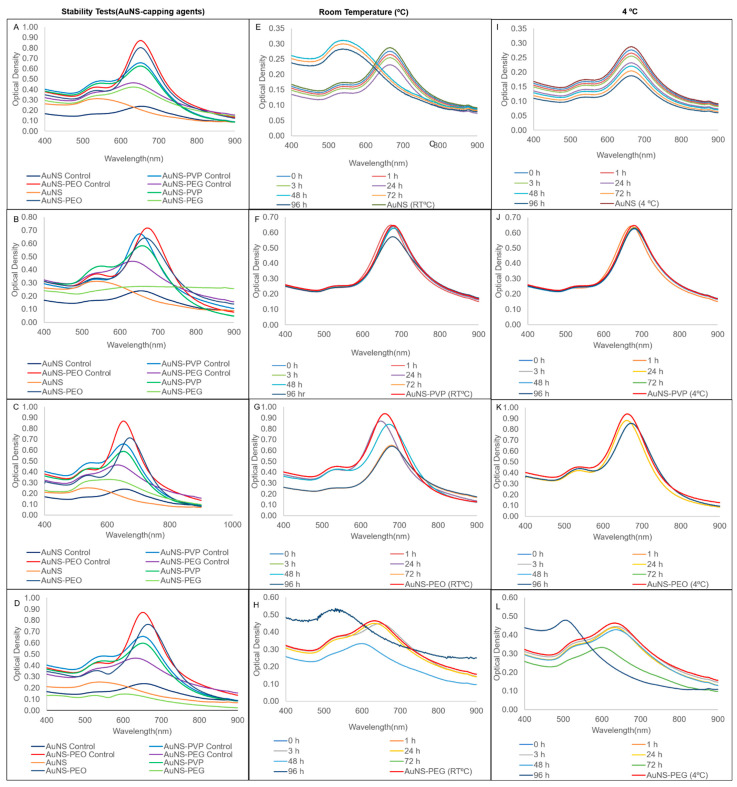
Comparison of UV-vis spectra of capped AuNS stability after 3 h of incubation in (**A**) salt; (**B**) urine; (**C**) serum; and (**D**) supplemented cell culture media. Storage stability UV-vis spectra at room temperature and 4 °C, respectively, for (**E**,**I**) AuNS; (**F**,**J**) AuNS-PVP; (**G**,**K**) AuNS-PEO; and (**H**,**L**) AuNS-PEG.

**Figure 4 nanomaterials-12-02470-f004:**
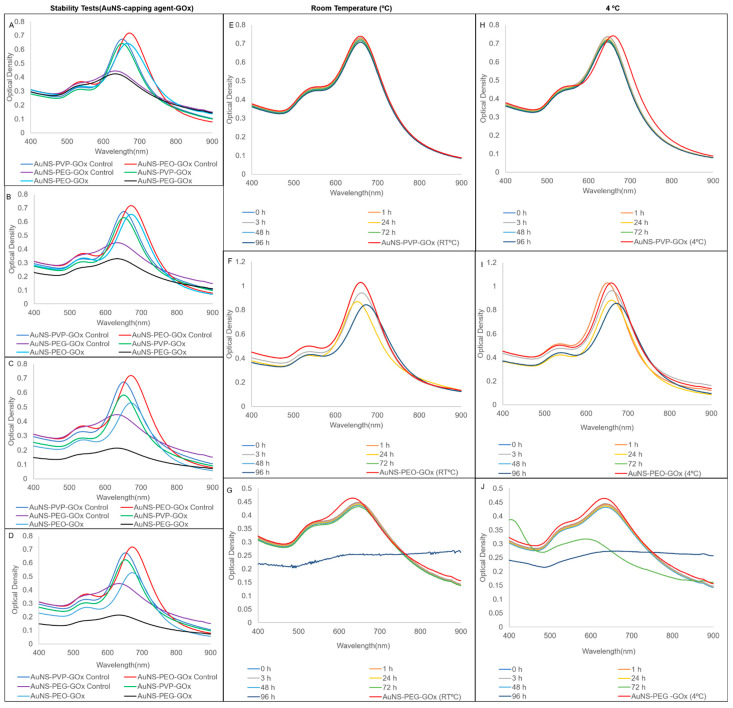
UV-vis spectra of stability comparisons for AuNSs bioconjugated with glucose oxidase after 3 h of incubation in (**A**) salt; (**B**) urine; (**C**) serum; and (**D**) supplemented cell culture media. Storage stability UV-vis spectra at room temperature and 4 °C, respectively, for (**E**,**H**) AuNS-PVP-GOx; (**F**,**I**) AuNS-PEO-GOx; and (**G**,**J**) AuNS-PEG-GOx.

**Figure 5 nanomaterials-12-02470-f005:**
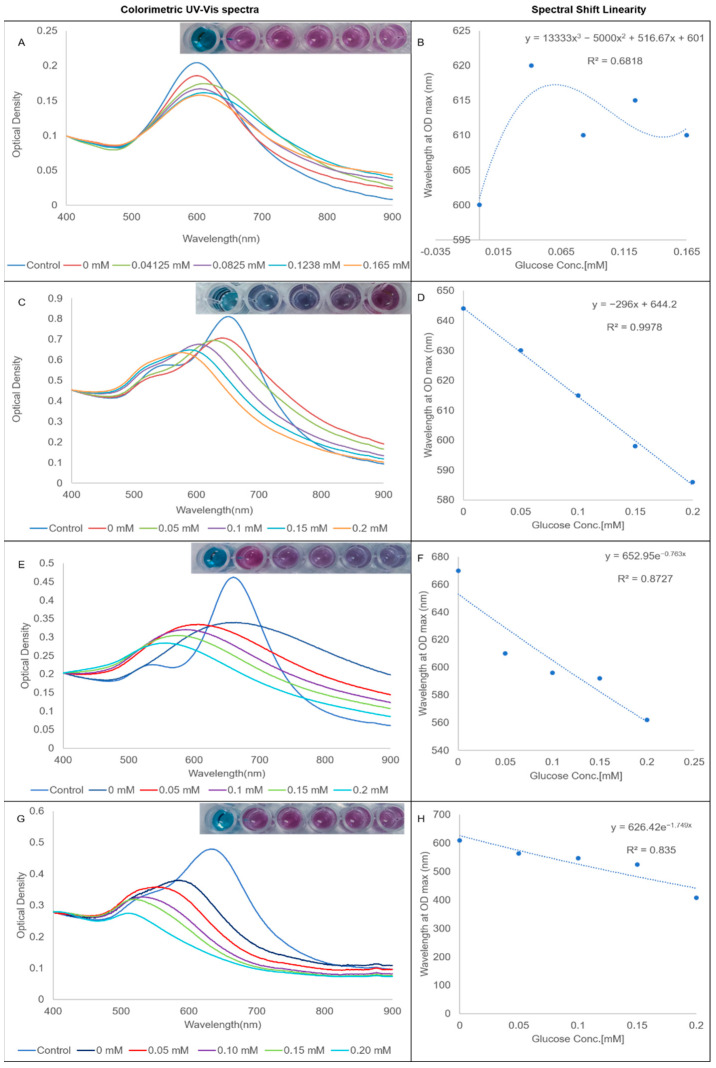
Glucose-sensing-efficacy-normalized UV-vis spectra and correlation curves: (**A**,**B**) AuNS-GOx; (**C**,**D**) AuNS-PVP-GOx; (**E**,**F**) AuNS-PEO-GOx; and (**G**,**H**) AuNS-PEG-GOx.

**Table 1 nanomaterials-12-02470-t001:** Summary of results: structure–function relationship.

Capping Agent	Structure	Side Group	Stability of Capped AuNS and Bioconjugates	Colourimetric Biosensing	Suggested Down Stream Applications
**PVP** **Mw:** **10,000**	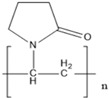 ${\left(C_{6}H_{9}NO\right)}_{n}$	Bulk Low segmental mobility Low hydrogen bond-forming capacity	**Short Term:** Excellent **Long Term:** Excellent	5 min response time Distinguishable colours by the naked eye Steady reaction	Colorimetric biosensing
**PEO** **Mw:** **100,000**	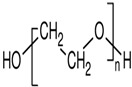 ${\left(-CH_{2}CH_{2}O-\right)}_{n}$	Linear High segmental mobility High hydrogen bond-forming capacity	**Short Term:** Moderate **Long Term:** Poor	Immediate response time Indistinguishable colours by the naked eye Unstable reaction	Confirmatory narcotics-drugs testing Confirmatory rapid test kit
**PEG** **Mw:** **8000**	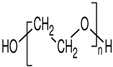 $H{\left(OCH_{2}CH_{2}\right)}_{n}OH$	Linear High segmental mobility High hydrogen bond-forming capacity	**Short Term:** Moderate **Long Term:** Poor	Immediate response time Indistinguishable colours by the naked eye Unstable reaction	Confirmatory narcotics-drugs testing Confirmatory rapid test kit

## Data Availability

Publicly archived datasets analyzed or generated during the study can be downloaded from DOI 10.6084/m9.figshare.20072186.

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
