# Peer review of "The Effect of Capping Agents on Gold Nanostar Stability, Functionalization, and Colorimetric Biosensing Capability"

_nanomaterials, 2022, doi:10.3390/nano12142470_

Round 1
Reviewer 1 Report
The work is interesting and can be accepted after minor revision.
1. Please improve English.
2. There are various formatting and typos errors.
3. There are relevant articles (Electrochim. Acta 252 (2017) 549-557.; Mater. Adv. 1 (2020) 2003-2009; Nanoscale Adv. 1 (2019) 719–727; J. Environ. Chem. Eng. 7 (2019) 103347) which should be cited in the introduction part.
4. Revise conclusion as your findings.
5. Revise Figure 1c and Figure 1f. The text is not visible.
Reviewer 2 Report
An explanation should be provided in terms of chemical structure/MW of capping agents on the different characteristics and behavior of AuNS PVP capped with respect to the AuNS PEG and PEO capped ones. The authors should try to discuss the results (the observed differences between the analyzed systems) in terms of structure-function relationship.
PEG and PEO have the same chemical structure, only their MWs are different (PEG: 8000 Da, PEO: 100000 Da, as reported by the authors).
PVP has a different chemical structure and a MW similar to PEG (PVP: 10000 Da, as reported).
This would be important for predicting the use of AuNS in specific applications, such as the colorimetric biosensors and would add a degree of novelty to the state of art.
Reviewer 3 Report
This paper reports how capping agents play a key role in altering the surface charge furthermore acting as bioconjugation scaffolds and the choice of synthesis method is pivotal to the final properties of the AuNS.
There are some of the questions below should be addressed.
1. The abstract seems to be unclear and incomplete. (What was found was that the choice of synthesis method is pivotal to the final properties of the AuNS including choice of capping agent, stability and downstream applications. Furthermore, the complex interaction between nanoparticles, capping agents and conjugated enzymes should be further studied). I suggest to change this paragraph with something of the form: In this study we proved that...
2. The quality of the figures is not sufficient. The numbers and other information from figure cannot be understood.
Round 2
Reviewer 2 Report
Thank you for the answer and the integration provided. The revised paper is suitable for publication.